# The Influence of Distinct Seasons on the Succession and Diversity of Bacteria on the Anticorrosive Coatings Surfaces in a Marine Environment

**Shuangwei Li, Jie Liu, Qian Li, Wenfang Li, Xinfeng Xiao and Linlin Zhang ***

College of Safety & Environmental Engineering, Shandong University of Science & Technology, Qingdao 266510, China
* Correspondence: linlinzhsd@126.com

**Abstract:** Epoxy resin has been frequently used as a coating paint for anticorrosion protection because of its excellent chemical properties. However, the long-term succession of bacteria colonizing coatings surfaces in the different seasons of the year remains uncharacterized. In this work, amplicon-based 16s rDNA sequencing was used to characterize the tempol change of bacterial communities growing on the epoxy resin surfaces. The results showed that bacterial diversity indices on spring and autumn immersion samples were higher than that of the samples immersed on summer and winter samples. Proteobacteria was found to be the dominant bacteria of all different seasons and accounted for 57.9% of the total sequence. *Gammaproteobacteria* and *Alphaproteobacteria* were the dominant classes in all of the samples, whereas the most abundance bacteria at the genus level had the significant differences with a change of season. Firmicutes also displayed a distinct temporal change pattern in that it was the second abundance in the summer and autumn samples, but had a marked decrease in the other season samples. These results demonstrated that bacterial community composition underwent obvious changes over the distinct seasons of a year. This study will be helpful for the seasonal change of bacterial diversity and development of corrosion-resistant paints.

**Keywords:** season change; temporal succession; bacterial community composition; 16S rRNA gene sequencing; anticorrosive coatings

## 1. Introduction

Metal corrosion is a worldwide concern that affects diverse productive industries, and the annual corrosion cost accounted for 3.34% of the gross domestic product (GDP) in China [1]. The marine environment is a highly corrosive and biologically active habitat for metals and causes severe destruction of infrastructure and industrial equipment. Corrosion is a complex phenomenon that occurs naturally and involves physical, chemical and biological processes [2]. Submerged metallic surfaces in the marine environment are rapidly colonized by microorganisms and formed biofilm, which provides numerous ecological advantages to microorganisms to degrade metal, causing severe structural damage, such as pitting and crevice [3]. Hence, the microorganism-induced corrosion process, known as microbiologically influenced corrosion (MIC), exerted an essential induction or acceleration factor in metals corrosion [4]. Currently, MIC has been extensively studied over a long period and at least 20% of steel corrosion in the marine environment is caused by microorganisms [5].

MIC is commonly associated with the participation of diverse microorganisms present on metal surfaces, especially bacteria. According to the demands for oxygen, the bacteria in corrosive environments was classified into three different categories of aerobic, anaerobic and facultative bacteria [6]. Although sulfate-reducing bacteria (SRB) belonging to the members of anaerobic bacteria in the biocorrosion have been well studied, other microbial groups also play a key role in the corrosion [7]. Various bacteria in a biofilm act

synergistically and cause more severe corrosion than if only one species, but most studies on biofilm-mediated corrosion focus on single species in the laboratory [8–10]. Hence, information on the identity of bacterial communities in corrosive environments is required, which is helpful to better understand the complex interactions between them [11].

Metal corrosion can be prevented by organic coatings which have been widely used under harsh environmental conditions for many years [12]. To further enhance the corrosion resistance of coatings under different marine conditions, resins and curing agents were applied on these coatings. Epoxy resin, a thermosetting polyepoxide polymer, is one of the most used coatings because of its excellent resistance to moisture, scratch hardness and chemical species along with good adhesion [13–16]. It was reported that different materials with corresponding physicochemical properties significantly influence the ability of microorganisms to adhere to a particular abiotic surface. After immersion in the seawater for a week, Proteobacteria, Actinobacteria, Firmicutes and Bacteroidetes were the main bacterial groups on the copper-based antifouling surface [17]. The main group was *Gammaproteobacteria*, with 93.9% distributed among 88 different representatives on 316 L stainless steel coupons after 30 days of seawater incubation [3]. However, most of the marine bacteria diversity studies published on artificial surfaces were rarely involved in anticorrosion coatings. In addition to the physicochemical properties of solid surfaces, environmental factors, such as pH, nutrients, salinity and temperature, play important roles in the biofilm formation and microbial surface colonization [18–21]. Temporal variation is another nonnegligible factor and important to determine the correct application time frame and for the selection of antifouling or anticorrosion paints [20,22]. For example, *Alphaproteobacteria* was found to be the dominant group on the days of biofilm growth in winter (day 1 to day 4), but marine microbial biofilm was dominated by *Gammaproteobacteria* and *Mollicutes* during the same days of biofilm growth in spring [20]. However, the impact of entire seasonal variations on the microbial communities adhering to the anticorrosion coatings has not been thoroughly analyzed.

Previous studies suggested that traditional cultivation methods lack sufficient sequences to obtain systematic and comprehensive information of microbial communities [23]. Due to modern technological developments, bacterial community structures in the material surfaces can now be analyzed using high-throughput sequencing technologies. Herein, the temporal succession of the bacterial community attendant anticorrosion coating coupons of epoxy resin was conducted by exposing these coupons to the natural seawater over the entire seasons of a year. For this, 16S rRNA gene sequencing was employed to analyze the dynamics of the bacterial community, and the results of this study will be helpful for the study of anticorrosion materials.

## 2. Materials and Methods

### 2.1. Study Site and Sampling

The Port of Qingdao, situated in Shandong Province, China, was selected for field assays. The steel plates used in this study had the following composition (wt %): C: 0.16, Si: 0.12, Mn: 0.45, S: 0.029, and P: 0.019. Carbon steel plate was used as the base substrate and cleaned with anhydride ethanol. Plates were then coated with each of two layers of the coating by paint brushes. Each plate comprised an epoxy bottom and a urethane top layer with thicknesses of 80 and 100 μm, respectively. All of the samples were fixed on the nylon ropes with metal chains as weights and then immersed below 3.0 m of the seawater. Samples were collected every three months (i.e., in March, June, September and December), and a total of 16 samples were collected for further analysis. The average temperature of the seawater was 13.7 °C and the pH was 8.3.

### 2.2. Samples Collection and DNA Extraction

All collected samples were quickly removed from the sea, and the large fouling organisms were removed with sterile forceps under sterile conditions. Coupon samples were rinsed with cleaned seawater to remove unattached bacteria and transported to

the laboratory for bacterial analysis. They were immediately placed in 40 mL sterile plastic centrifuge tubes, transported to the laboratory on dry ice, and were stored at $-80\,^\circ$C pending analysis. The total DNA genomic was extracted using a PowerSoil® DNA Isolation Kit (MOBIO Laboratories, Carlsbad, CA, USA), according to the manufacturer. The extracted DNA concentration and purity was determined by Ultramicro spectrophotometer (Nanodrop 2000, Thermo Scientific, Wilmington, NC, USA).

### 2.3. Processing Sequencing Data

Obtained DNA samples were sent to Novogene Bioinformatics Technology Co., Ltd. (Beijing, China) for bacterial community analysis by using high-throughput pyrosequencing based on an Illumina platform. Amplicon libraries were constructed for pyrosequencing using primers 341F (5′-CCTACGGGRSGCAGCAG-3′) and 806R (5′-GGACTACHVGGGTWTCTAAT-3′) targeting the 16S rRNA V3/V4 region [24,25]. The obtained sequence data were processed with Cutadapt v1.10 (TU Dortmund, Germany) to shear quality part reads. Barcode and primer sequences were cut off and raw reads were obtained by preliminary quality control. Then, the final valid data (clean sequences) were obtained by demultiplexing, trimming the primer sequence, quality filtering and correction of sequencing errors. Clean sequences with ≥97% similarity were assigned to the same operational taxonomic unit (OTU) using Uparse software (California, CA, USA). The resulting OTUs were used in all subsequent analyses. Then, species annotation was performed for OTUs sequences using the Mothur and SSUrRNA database of SILVA132 (MA, USA). The community composition was analyzed and multiple sequence alignment was executed with MUSCLE 5 software (California, CA, USA).

### 2.4. Bioinformatics Analysis

Bioinformatics analysis was performed by Novogene Bioinformatics Technology Co., Ltd. (Beijing, China). In order to attribute taxonomy, only sequences with 97% of identity hits in an alignment covering over 99% were considered. OTU data were used for alpha-diversity analysis, including Simpson, Shannon, Chao and dominance, and displayed with the vegan R package (V2.15.3) (Helsinki, Finland). The phylogenetic relationship was inferred using the neighbor-joining method. The evolutionary distances were computed using the maximum composite likelihood method, and evolutionary analyses were conducted in MEGA X (Philadelphia, PA, USA). The stacked bars graphs, Venn diagram and heatmap were made using the ggplot2 and Vegan packagesavailable for the R Studio software (Helsinki, Finland). The Shannon index, Simpson diversity index, Chao1 and observed species of each sample were used to evaluate the species richness, and diversity indicates a large distinction of species richness and diversity of the bacterial community within the sample. To further compare the community differences among these samples, beta diversity employing clustering analysis and Non-Metric Multidimensional Scaling (NMDS) were conducted and plotted.

### 2.5. Statistical Analysis

The results analysis was carried out using SPSS 22 (IBM, Chicago, IL, USA) to evaluate the effect of ivermectin on the antifouling performance of the coatings. It was considered significant if $p < 0.05$. To compare the order, class and genus abundance of different biofilm groups, one-way ANOVA was employed using values from the biological biofilm replicates.

## 3. Results

### 3.1. Diversity Analysis of Bacteria in Different Seasons

The bacterial diversity present in the coupons covered by the anticorrosion paint was investigated during four time points of season by rRNA 16S sequencing. For the alpha-diversity analysis, Shannon and Simpson indices, and non-parametric richness estimators (Chao1 and ACE) were used in the samples evaluated (Table S1). The heterogeneity indices of the Shannon and Simpson measures indicated that the autumn sample library had

the highest values with 7.40, while the spring sample had a smaller diversity of 6.16, which was larger than the summer samples with 5.6 (Figure 1A,B). The lowest values were found in the winter sample library with an index of 4.1. Additionally, the indices of abundance calculated by Chao1 and ACE displayed that the results of the samples of spring and autumn both had a higher value than other samples, then followed by the summer and winter samples library in the abundance index (Figure 1C,D). The richness differences among these community structures of coupons were shown in the rarefaction curve analysis and exhibited the same characteristics as the previous results. Figure 1 shows that the autumn sample had the greatest richness, while conditions at summer and winter showed a similar index (Figure 2A). It can be observed that the structures of the bacterial communities from four different seasons using NMDS indicated a distinct separation among these samples (Figure 2B).

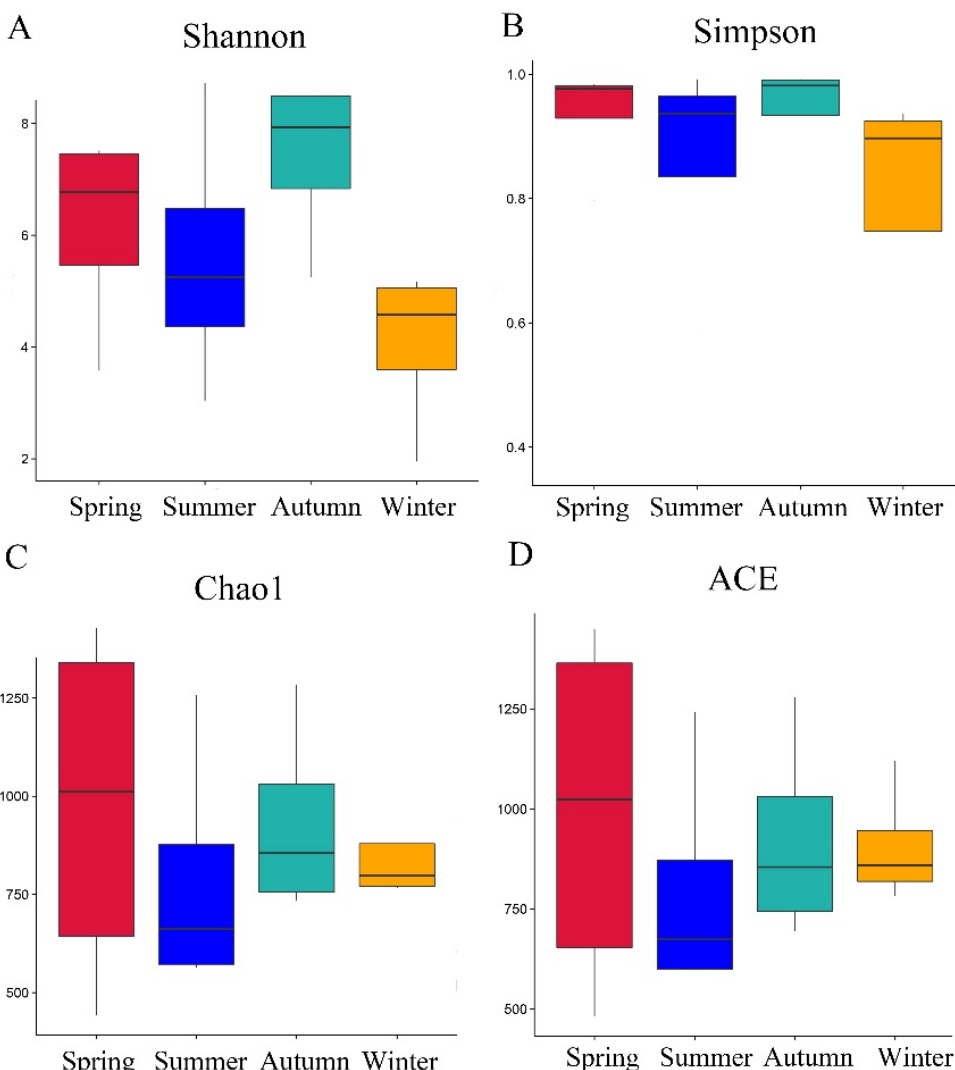

**Figure 1.** Diversity and community richness indices in four different seasonal groups. (**A**) comparison of Chao1 diversity indices in four different seasonal samples. (**B**) comparison of Shannon diversity indices in four different seasonal samples. (**C**) abundance measurement index calculated by the Simpson method. (**D**) abundance measurement index was shown by the ACE.

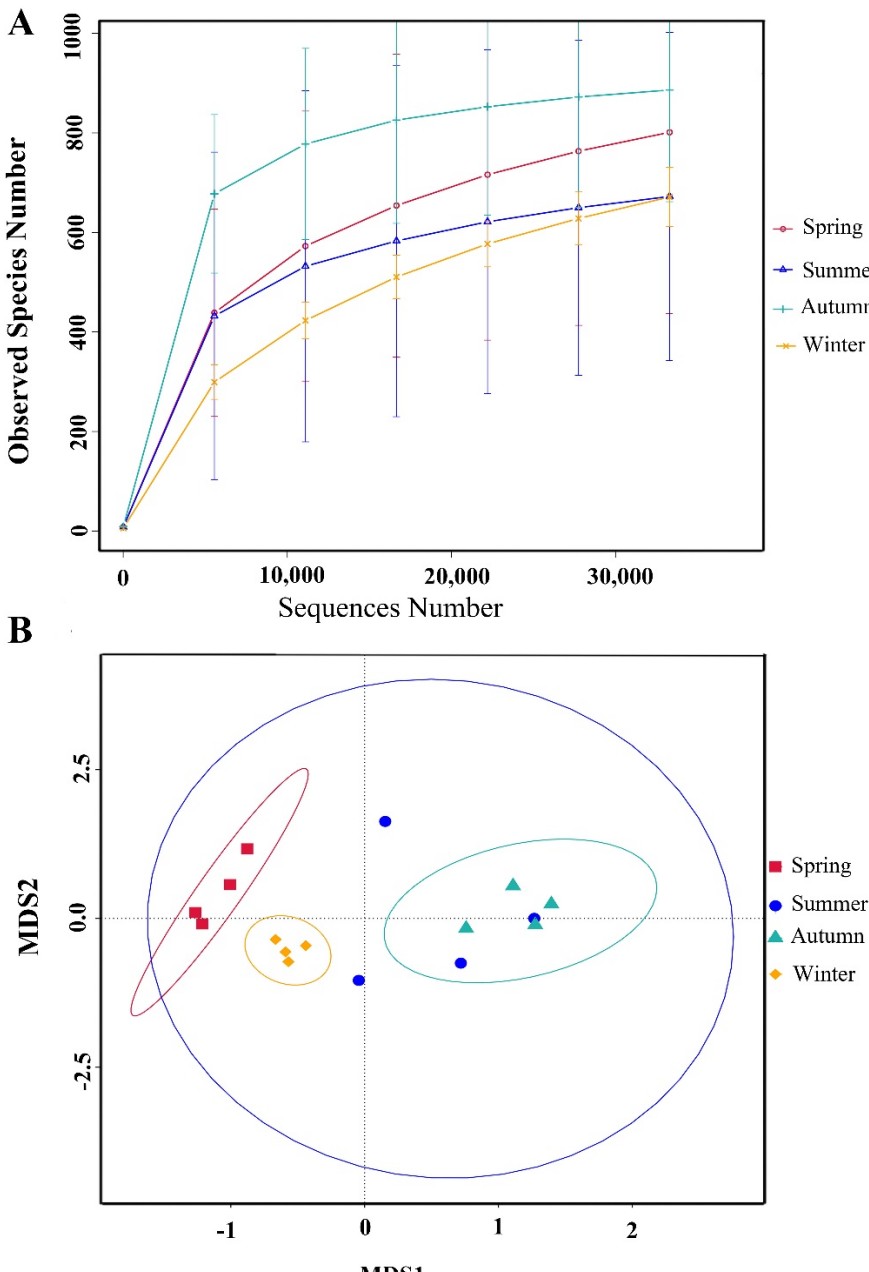

**Figure 2.** The rarefaction curve and NMDS analysis of all samples in different seasons. (**A**) Rarefaction curve analysis showing the relationship between sequences number (*x*-axis) and observed species number (*y*-axis). (**B**) NMDS plot of Bray–Curtis distances among four seasonal samples based on OTUs for bacteria.

OTUs distributions and alpha-diversity indices of sample libraries in all periods were compared and are presented in Table S1. A total of 7163 high-quality OTUs sequences distributed in 45 phyla, and 460 species were generated by the Illumina MiSeq (Beijing, China) sequencing at the four time points of season analyzed. After the exposure to the spring in the marine environment, 1639 OTUs were generated in the sequenced coupons. Afterwards, there was a rapid decrease in the number of OTUs at the time of the summer and 1990 OTUs were found from the collected coupons. Although a continuous increase of OTUs (2174 OTUs) in the autumn sample was found, there was a marked decrease in the number of sequences with 1360 OTUs in the winter samples. The distribution and taxonomic groups overlapping the identified OTUs in the different seasonal samples are presented by a Venn diagram in Figure 3. Despite the temporal differences for bacterial

microorganisms, 169 shared OTUs were found in all of the samples. The number of unique OTUs (754) for the autumn sample was the largest, followed by spring and summer samples, respectively. Finally, the number of OTUs (177) present exclusively in the winter samples was the lowest.

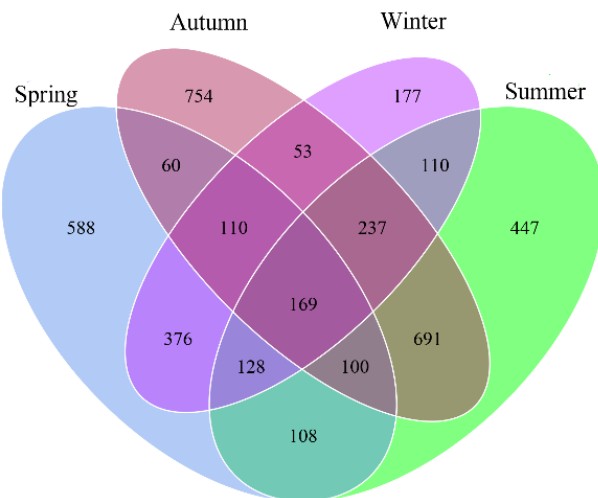

**Figure 3.** Venn diagram describing the OTU overlap in microbial communities analyzed in the different seasons.

### 3.2. Community Diversity Analysis

In the spring sample, obtained OTUs were distributed in 13 different phyla taxonomic groups, and the most predominant bacteria were proteobacteria, which constituted 52.2% of the total detected sequences (Table S2). Proteobacteria contained 290 species distributing in three classes with 83 different families. The second abundant group was the unidentified bacteria, then followed by Bacteroidetes comprising 19.9% of all the sequences. The phylum of Latescibacteria was detected only at this time point of analysis. As for the class-level sequencing, the massive representation was identified as *Gammaproteobacteria* (29.9%). Furthermore, the most representative genus on the community composition of the *Gammaproteobacteria* was *Vibro*, *Colwellia*, *Marinomonas*, *Pseudoalteromonas*, *Oleiphilus*, *Halioglobus*, *Alkalimarinus* and *Psychromonas*. *Bacteroidia* was the second-largest class in the spring samples analysis, with 326 OTUs and nine different representative genera were detected. *Alphaproteobacteria* appeared in third place, with 17.5% of all sequence numbers, and the most abundant genus in this class mainly referred to *Sulfitobacter*, *Loktanella*, *Roseovarius* and *Litoreibacter*.

During the summer analysis period, a total of 20 different phyla were found and identified on the collected coupons (Table S2). As the main group on the phylum level, Proteobacteria with 1039 OTUs occupied about half of all the sequences of samples. The second-most prevalent group was Firmicutes, with 370 OTUs (18.6%) and 14 different family members. Bacteroidetes was the third-most prevalent group on the bacterial abundance, presenting 326 OTUs (16.4%) and 25 different family members (Figure 4). The largest number at the class level was *Gammaproteobacteria* (40.7%), which had 48 different species members. *Stenotrophomonas* belonging to *Xanthomonadaceae* family represented more than 20% of all detected sequences and was the most abundant representative species. *Moraxella* and *Actinobacillus*, with 7.11% and 4.87% in all sequences, respectively, both affiliated to *Gammaproteobacteria*, were detected. It can be seen that the bacteria composition of the *Gammaproteobacteria* class was obviously distinguished from the spring samples. Additionally, only the *Zetaproteobacteria* were found at this time point of analysis. The *Bacteroidia* cluster was the second-most abundant in class with 15.0% and had the representativeness with *Marinifilum* and *Roseimarinus*, from *Marinifilaceae* and *Prolixibacteraceae* family, respectively. Among the major genus representatives of the *Clostridia* class, they were identified

as *Fusibacter* with a 5.0% rate of all sequences, respectively. The *Streptococcus* genus from the *Bacilli* class was detected and represented with a 2.6% score rate of all sequences.

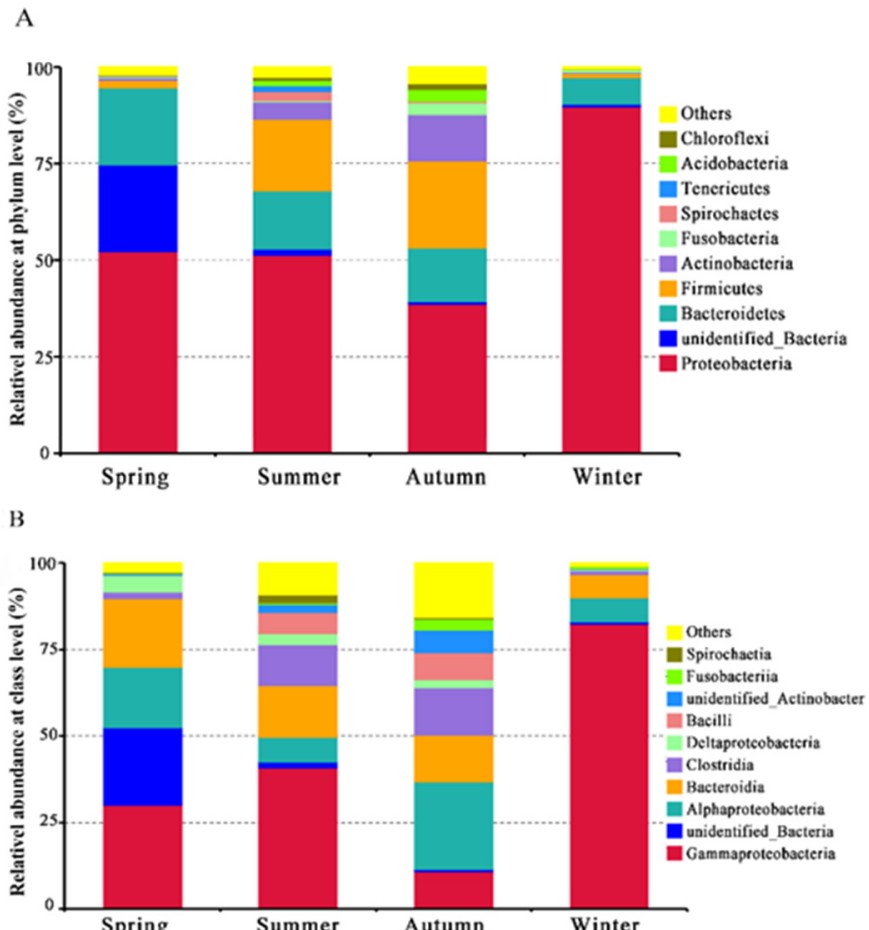

**Figure 4.** Comparison of microbial community structures on the different seasonal groups. (**A**) microbial community compositions at the phylum level. (**B**) microbial community compositions at class level.

When the autumn samples data were analyzed in the phylum taxonomic level, a total of 23 different phyla groups were obtained based on the sequencing results (Table S2). Although Proteobacteria were still determined as the main cluster presented on steel coupon covers, its proportion has dropped to 38.6% compared to that with the previous seasons. On the contrary, the proportion of Firmicutes in all the sequences continuously rises to 22.6% with OTUs distributed on the four classes. Meanwhile, the proportion of Actinobacteria also rises from 4.43% in summer to 12.0% in autumn. On the class level, the *Alphaproteobacteria* had a rapid rising and became the main group represented instead of *Gammaproteobacteria*. *Alphaproteobacteria* had 552 OTUs and was divided into 72 different specie members occupying 25.4% representativeness of all sequences. *Gammaproteobacteria* declined to the second place with a rate of 10.7%, followed by *Bacteroidia* and *Clostridia*, which had a similar rate of 13.7% and 13.6% between them. There was an order predominance of *Clostridiales*, with 296 of the OTUs in the *Clostridia* class. *Bacteroidales*, with the highest OTU number of 161, was the main group represented in the *Bacteroidia* and had 7.4% of all sequences. Finally, as the typical taxonomic representative in the Firmicutes, *Bacilli* had the highest OTU number and represented 7.7% of the identified sequence of all obtained sequences (Figure 5).

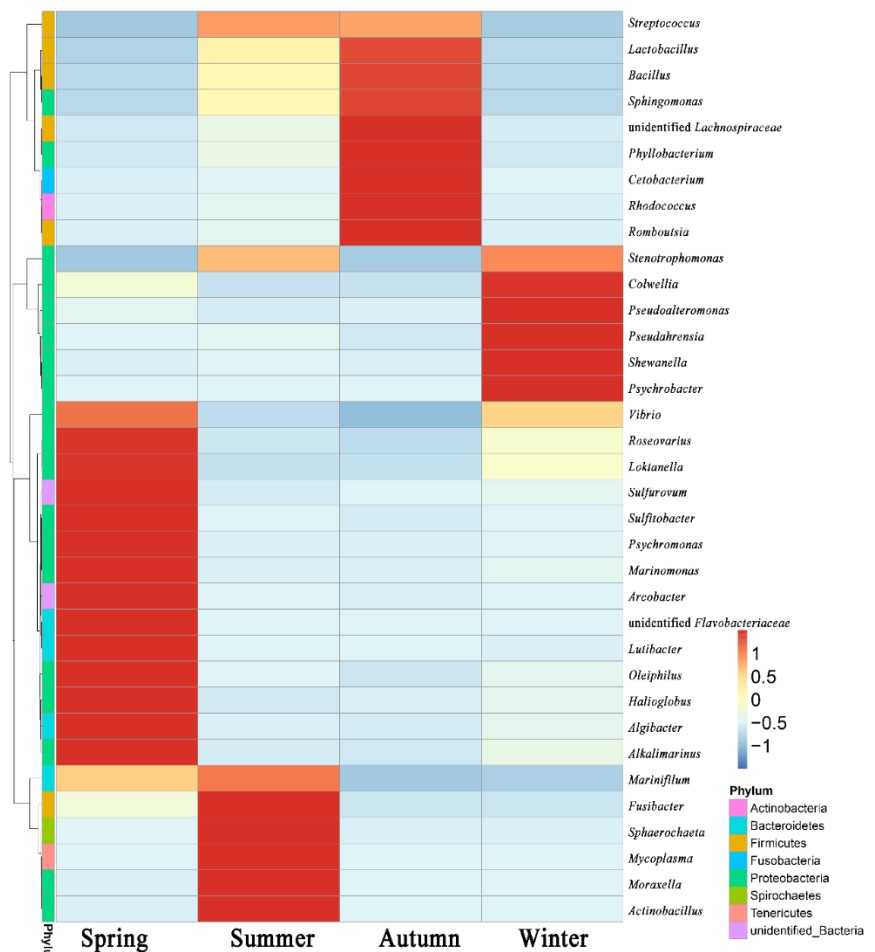

**Figure 5.** Comparison of the microbial community structures at the genus level on the different seasonal groups.

Within the winter incubation, it can be observed that the diversity of bacterial composition was greatly diminished. Although Proteobacteria occupied up to 89.7% in all the sequences and continued as the most representative bacteria group, the proportion of other phyla bacteria, like Firmicutes and Bacteroidetes, had a dramatic rate decrease. Bacteroidetes was the second-most abundant and only represented 13.9% of all detected sequences, followed by Actinobacteria with only 12.0% (Figure 4). As for the class, the most predominant bacteria were overwhelmingly dominated by *Gammaproteobacteria*, with 82.5%. Among the most abundant genus, *Stenotrophomonas*, belonging to the *Xanthomonadaceae* family, had a close score with the genus *Pseudoalteromonas*, belonging to the *Pseudoalteromonadaceae* family.

The main representative orders for *Alphaproteobacteria* were *Rhodobacterales* and *Rhizobiales*, with 2.6% and 2.0% of sequence numbers, respectively. The *Bacteroidia* were represented by *Flavobacteriales*, which had 4.4% of all sequence numbers. *Pseudoalteromonas* and *Colwellia* belonging to *Gammaproteobacteria* were found with 25.1% and 17.5% of all sequences, respectively. Additionally, there were similar sequences between *Alphaproteobacteria* and *Bacteroidia*, only with 3.02% and 1.89%, respectively.

## 4. Discussion

It has been frequently reported that the presence of microorganisms over material surfaces made it possible to induce or accelerate the metallic corrosion on the marine condition. To prevent the biocorrosion of steel, epoxy resin has been used as one of the most-used corrosion-resistive coating materials. The new studies have increasingly

suggested that the diversity and composition of the bacteria community plays an important role in the metal process and could be influenced by different environmental factors or seasons, subsequently changing the anticorrosion performance of materials. Although distinct temporal succession of bacterial communities in stainless steel surfaces has been already described in the scientific literatures, whether seasonal change had an effect on the bacterial composition covering on the epoxy resin needs to be further explored [15,16]. Therefore, the abundance and diversity indexes were employed to analyze the temporal succession within a microbial community on the surface of corrosion-resistive coating materials in this survey.

Herein, Proteobacteria phylum was dominant in all the samples at a range from about 38% to 90%, which was similar to the other studies reported [26]. As the pioneer surface colonizers, Proteobacteria has been widely described as the main taxonomic group across the different seasons and environmental conditions [17,20,27]. On the class level, *Gammaproteobacteria*, *Deltaproteobacteria* and *Alphaproteobacteria* were detected in all the samples, but only *Zetaproteobacteria* was found in the spring sample. When comparing the distribution of these classes among different seasonal samples, *Gammaproteobacteria* showed an obvious dominance in all the samples except the autumn sample. The highest proportion of *Gammaproteobacteria* was found in the winter sample with about 90% of the OTUs, but significantly decreased to 10.7% in the autumn sample. In contrast, the abundance of *Alphaproteobacteria* increased to 25.4% in the autumn sample and the lowest was detected in the winter sample with only 6.8%. The *Gammaproteobacteria* and *Alphaproteobacteria* class relating to the corrosion process in metals has been described in the coupons immersing in the marine seawater [27]. Previous study showed that carbon steel disks biofilms were dominated by *Gammaproteobacteria* and *Alphaproteobacteria* comprising $58 \pm 19\%$ and $34 \pm 16\%$ of sequences on average, respectively [28]. Among the representatives of *Gammaproteobacteria*, the predominant order *Alteromonadales* with the representative genus of *Colwellia*, *Oceanospirillales* and *Pseudoalteromonas* occupied a considerable proportion in the winter sample. Another order recognized constantly was *Xanthomonadales* having the representative genus of *Stenotrophomonas*, which was the most-abundant genus in the summer sample. The involvement of these bacteria in the metal corrosion occurring in the marine corrosive steel has been described [11,26]. *Colwellia* was commonly found in the cold marine environment and also known to degrade hydrocarbons [29]. *Stenotrophomonas* isolating a wide variety of environments has confirmed sulfate-reducing behavior or contributed to the formation of biofilm by excreting exopolysaccharides, which can facilitate the attachment of other microorganisms [30]. *Pseudoalteromonas* species not only had the ability to cause corrosion of 2205 duplex stainless steel alloys adhesion and the formation of biofilm, but also can accelerate the corrosion of low alloy steel by the endogenous electron mediator pyomelanin [31,32]. However, the research on these bacteria still needs to be explored in the natural marine environment. The iron-oxidizing bacteria *Zetaproteobacteria*, found in the shallow environments with a relative low abundance, reportedly was observed that it may be part of successional colonization leading to MIC [33]. Another research reported that *Zetaproteobacteria* was found in a single-control biofilm sample in low abundance (0.06% OTUs), which was similar with this research.

Bacteriodetes, the common phylum in the biofilms, was the second place of the total samples, but its representativeness was much lower when compared to Proteobacteria. The diversity of Bacteroidetes involved in metal bio-corroding microbiota in seawater environments have been detected according to other recent metagenomic reports [34,35]. The range of Bacteriodetes was from 6.9% to 19.9% of the total number of phyla and the highest abundance was found in spring, while the lowest was in winter. To analyze the main classes belonging to the Bacteriodetes phylum, it can be found that *Bacteroidia* was predominant in this study. *Bacteroidia* was mainly distributed between two kinds of *Bacteroidales* and *Flavobacteriales* orders. Nonhydrogenotrophic nitrate-reducing bacteria *Prolixibacter* belonging to *Bacteroidales* corroded FeO concomitantly with nitrate reduction, and the amount of iron dissolved by the strain was six times higher than that in an aseptic control [36].

Moura et al. reported that *Flavobacteriales* are efficient surface colonizers participating in the steel coupons corrosion during the first 15 days in the microcosm experiments and can use chemolithotrophic Fe- and S-oxidizing bacteria to take the advantage of primary production during biofilm formation [7,37]. Interestingly, about 22.5% of species were identified as unknown; this might be due to the fact that some of the strains were unidentified so far, and their role in the biofilm formation and acceleration of corrosion reactions need to be explore in the future.

Firmicutes phylum OTUs corresponded to about 11% of the total, whereas its abundance distribution on the season has a marked difference. The abundance of Firmicutes in the summer and autumn, with 18.6% and 22.6%, respectively, was much higher than the other two seasons of spring and winter, with only 2.0% and 1.1%, respectively. The presence of Firmicutes in the study of metal coupons biocorrosion within marine water incubation was commonly reported [20,23]. When comparing the distribution of Firmicutes bacteria on the class level, the dominance class was *Bacilli* and *Cytophagia* with about 3.5% and 7.0% of the total OTUs, respectively. The *Bacilli* class previously was detected in the experiment and suggested its participation in influencing microbiological succession in carbon steel corrosion [3]. The representative generous of *Bacilli* was *Bacillus*, which was one of the dominant culturable chemo-organotrophic bacteria from estuarine and marine environments in other works [38]. Although *Bacillus* was not the main participator in corrosion of metals in this study, its presence in biocorrosion studies has been commonly described [26,39]. *Bacillus* was associated with MIC by degrading a wide range of aliphatic and aromatic hydrocarbon and act as a sole carbon source under the marine environments [40]. Another dominance genus of the *Bacilli* was *Streptococcus* which induced severe pitting corrosion on the 316 L stainless steel under an anaerobic environment with high sucrose by producing organic acids [41]. Interestingly, *Bacillus* and *Streptococcus* was only found on the summer and autumn samples.

Actinobacteria with 4.3% of total OTUs especially had the highest rate of 12.0%, which was a nonnegligible proportion in the autumn sample. Actinobacteria has been found on the carbon steel and copper-based antifouling paint, and has been associated with corrosion [17,26]. Finally, as for other bacterial phyla, such as Acidobacteria, Nitrospirae and Fusobacteria, were also detected on the coupons, but below 2% in considerably low numbers of OTUs.

## 5. Conclusions

This work investigated the long-term seasonal succession spanning an entire year of taxonomic profiles and diversity of the bacterial communities attached to the anti-corrosive coating with epoxy resin. The result demonstrated that the bacterial diversity from spring and autumn samples was significantly higher than that taken from samples immersed in the summer and winter seasons. Proteobacteria was the dominant bacteria among the different seasons; its highest abundance appeared in winter and lowest was in the autumn. However, the notable differences in microbial compositions on the genus level were observed at different seasons. Bacteria with the highest abundance on the genus level from the previous season was almost completely replaced in the next season, suggesting that bacterial community composition has undergo significant variations even the sample location was unchanged. Additionally, Firmicutes and Bacteroidetes associated to the corrosion of the metals were also found in the different seasonal samples, with a much lower abundance than Proteobacteria, but their abundance was also dependent on the sampling season. These results revealed the dynamic change nature of these bacteria under the different season. Therefore, the temporal succession of marine bacteria communities should be taken in consideration in the future anticorrosive applications.

**Supplementary Materials:** The following supporting information can be downloaded at: https://www.mdpi.com/article/10.3390/w14193183/s1, Table S1: Numbers of OTUs and taxa, diversity and community richness indices in different seasons; Table S2: Microbial community compositions at phylum level; Table S3: Microbial community compositions at class level.

**Author Contributions:** S.L.: Writing original draft preparation. J.L. and W.L.: Experiment performed and Methodology. Q.L.: Data analysis. X.X.: Formal analysis. L.Z.: Writing—reviewing and editing. All authors have read and agreed to the published version of the manuscript.

**Funding:** This research received no external funding.

**Data Availability Statement:** Not applicable.

**Acknowledgments:** We are grateful to the anonymous reviewers for valuable comments and the editors for careful editing.

**Conflicts of Interest:** The authors declare no conflict of interest.

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
