# Peer review of "The Influence of Distinct Seasons on the Succession and Diversity of Bacteria on the Anticorrosive Coatings Surfaces in a Marine Environment"

_water, doi:10.3390/w14193183_

Round 1

Reviewer 1 Report

The article provides a good overview on seasonal changes of microorganisms detected in Qingdao harbor. It shows the variety of bacteria grown at certain seasons of just one year. No judgement is made in terms of long term situation (i.e. different years).

However, no influence on the coating was investigaed and reported. Also no effect of microorganisms found on the actual corrosion behavior of metals in marine environment compared to normal seawater influence is discussed.

I recommend to change title and scope in order to address the research correctly.

Additionally, I recommend checking language and use of correct terminology.

Some examples:

Title: "... in marine environment"

page 3, 143: ... coupons covered by protective coating...

Author Response

Dear reviewer,

We would like to thank you for your kindly help to revise this manuscript and consider our manuscript for publication in Water after major revisions. We appreciated all the comments and suggestions and carefully considered all of them during the revision. Here is a summary of revisions and responses.

The article provides a good overview on seasonal changes of microorganisms detected in Qingdao harbor. It shows the variety of bacteria grown at certain seasons of just one year. No judgement is made in terms of long term situation (i.e. different years).

However, no influence on the coating was investigated and reported. Also no effect of microorganisms found on the actual corrosion behavior of metals in marine environment compared to normal seawater influence is discussed.

I recommend to change title and scope in order to address the research correctly.

Authors’ responses: Thanks for your comments. The article title has been changed for better describing the seasonal changes of microorganisms on the anti-corrosive coatings surfaces.

Additionally, I recommend checking language and use of correct terminology.

Some examples:

1.Title: "... in marine environment"

Authors’ responses: Thanks for your comments and suggestions. The incorrected words have been revised in the new manuscript.

  1. page 3, 143: ... coupons covered by protective coating...

Authors’ responses: Thanks for the reviewer’s comments. This sentence has been corrected. Additionally, the language and terminology have been checked carefully and improved in the new manuscript.

We did our best to address your suggestions and comments during the revision. Hope the revised MS is acceptable for publication. We are looking forward to hearing your decision. Thank you very much for your help.

Best regards.

Linlin Zhang, Ph.D.

Email: linlinzhsd@126.com

Reviewer 2 Report

In the article presented for review, the research e results demonstrated that bacterial community composition underwent obvious changes over the distinct seasons of a year. This study will be helpful for the seasonal change of bacterial diversity and development of corrosion resistance paints. It is also very important research in the face of what is happening now with rivers and water reservoirs in Poland - the Odra River. These materials can be a source of secondary pollution.

The article is well written. However, I have a few comments:

           Please separate the captions under the pictures from the text.

           Figures should also be separated from the text, e.g. picture 5.

           Between lines 39 and 40, 50 and 51, and 74 and 75, you use more space, unnecessarily.

           Could figure 2 be clearer?

           Please check and correct the name of the microorganisms. Most of the time it's fine, but mistakes do happen.

Thank you for considering my opinion. I encourage the authors to continue working on improving the manuscript

Author Response

Dear Reviewer,

We would like to thank you for your thoughtful review. We have carefully edited the manuscript and have made many changes to both the text and figures in response to your comments, and we believe it will improve the overall quality of the manuscript. Below is our response to each of your comments:

In the article presented for review, the research e results demonstrated that bacterial community composition underwent obvious changes over the distinct seasons of a year. This study will be helpful for the seasonal change of bacterial diversity and development of corrosion resistance paints. It is also very important research in the face of what is happening now with rivers and water reservoirs in Poland - the Odra River. These materials can be a source of secondary pollution.

The article is well written. However, I have a few comments:

1.Please separate the captions under the pictures from the text.

Authors’ responses: Thanks for your comments and suggestions. The captions under the pictures have been separated from the text in the revised manuscript.

2.Figures should also be separated from the text, e.g. picture 5.

Authors’ responses: Thanks for the reviewer’s comments. Figures 5 would be separated from the text in the new manuscript.

3.Between lines 39 and 40, 50 and 51, and 74 and 75, you use more space, unnecessarily.

Authors’ responses: We appreciate the reviewer’s comments. The unnecessary spaces in these lines have been removed.

4.Could figure 2 be clearer?

Authors’ responses: We appreciate the reviewer’s comments. Figure 2 was replaced by the clearer figure.

5.Please check and correct the name of the microorganisms. Most of the time it's fine, but mistakes do happen.

Authors’ responses: We appreciate the reviewer’s suggestions. All of the names of the microorganisms were checked again and the inaccurate names were corrected.

6.Thank you for considering my opinion. I encourage the authors to continue working on improving the manuscript

Authors’ responses: Thanks for the reviewer’s comments. The manuscript has been improved and our work will be continued as you suggested.

On behalf of the other authors, I wish to thank you again. We did our best to address all of your suggestions and comments during the revision. Hope the revised MS is acceptable for you. We are looking forward to hearing your decision. Thank you very much for your help.

Best regards.

Linlin Zhang, Ph.D.

Email: linlinzhsd@126.com

Round 2

Reviewer 1 Report

Thanks for considering my comments. Now the title reflects the actual content of research. However, I would delete "the" in front of "anti-corrosive coating".